# Variation of Glucosinolate Contents in Clubroot-Resistant and -Susceptible *Brassica napus* Cultivars in Response to Virulence of *Plasmodiophora brassicae*

**DOI:** 10.3390/pathogens10050563

**Published:** 2021-05-06

**Authors:** Nazanin Zamani-Noor, Johann Hornbacher, Christel Joy Comel, Jutta Papenbrock

**Affiliations:** 1Federal Research Centre for Cultivated Plants, Institute for Plant Protection in Field Crops and Grassland, Julius Kühn-Institute (JKI), Messeweg 11-12, D-38104 Braunschweig, Germany; 2Institute of Botany, Leibniz University Hannover, Herrenhäuserstr. 2, D-30419 Hannover, Germany; J.Hornbacher@botanik.uni-hannover.de (J.H.); joycomel20@gmail.com (C.J.C.)

**Keywords:** oilseed rape, polygenic resistance, physiological race, pathotype, clubroot severity index, aliphatic, aromatic and indolic glucosinolates

## Abstract

The present study investigated the changes in total and individual glucosinolates (GSLs) in roots and leaves of different clubroot-resistant and -susceptible oilseed rape cultivars following artificial inoculation with *Plasmodiophora brassicae* isolates with different virulence. The results showed significant differences in clubroot incidence and severity as well as in the amount of total and individual glucosinolates between oilseed rape cultivars in response to virulence of the pathogen. Single among with total aliphatic and total indolic glucosinolate contents were significantly lower in leaves of susceptible cultivars compared to resistant ones due to the infection. Similarly, single and total aliphatic as well as indolic glucosinolate contents in roots were lower in susceptible cultivars compared to resistant cultivars analyzed. The different isolates of *P. brassicae* seem to differ in their ability to reduce gluconasturtiin contents in the host. The more aggressive isolate P1 (+) might be able to suppress gluconasturtiin synthesis of the host in a more pronounced manner compared to the isolate P1. A possible interaction of breakdown products of glucobrassicin with the auxin receptor transport inhibitor response 1 (TIR1) is hypothesized and its possible effects on auxin signaling in roots and leaves of resistant and susceptible cultivars is discussed. A potential interplay between aliphatic and indolic glucosinolates that might be involved in water homeostasis in resistant cultivars is explained.

## 1. Introduction

Glucosinolates (GSLs), a known group of constitutive natural plant metabolites, are found in the order Brassicales, mainly in the family of Brassicaceae but also in families of Capparaceae, Caricaceae, Resedaceae and Tropaeolaceae [1]. Based on their side-chain structure and amino acid precursors, GSLs are divided into three major groups: aliphatic, aromatic and indolic [2]. Upon tissue damage such as insect feeding or fungal infestation, GSLs are catabolized by myrosinases to produce a variety of bioactive compounds such as isothiocyanates, thiocyanates, nitriles, oxazolidenethiones and epithioalkanes [1]. Some of these metabolites have been discovered to be toxic to many insect herbivores and some fungal pathogens and play important roles in the plant’s defense mechanism [1,3].

*Plasmodiophora brassicae* Woronin, the causal agent of clubroot disease, is one of the most destructive and cosmopolitan of plant pathogens. This obligate soil-borne protist attacks over 3700 species of the family Brassicaceae [4] including economically important oilseed rape (*Brassica napus* L.) cultivars. Effective disease control strategies against *P. brassicae* continue to be a challenge because of the lack of chemical agents able to manage this disease. Further, cultural practices tend to reduce the severity of clubroot, but none directly control *P. brassicae* on its own. Host resistance offers the only economic and sustainable method for the adequate managing of clubroot disease. However, current oilseed rape cultivars relying on race-specific resistance [5] often lose effectiveness within a few years by imposing selection for virulent pathotypes. Physiological specialization has long been known to occur in *P. brassicae* [6], with pathotypes of the pathogen varying in their ability to infect specific host crops. Previous studies in European countries have revealed variations in pathotype distributions across different countries [7,8,9,10]. Pathotype 1 (P1) and Pathotype 3 (P3) or ECD 16/31/31 and 16/14/31 as classified on the differentials of Somé et al. [11] and the European Clubroot Differential [12], respectively, are predominant in central Europe. Additionally, several *P. brassicae* populations were found to be moderately or highly virulent on currently available clubroot-resistant oilseed rape cultivars [7,8,9,10,13]. These new isolates have been informally named as P1 (+), P2 (+) or P3 (+) because they are classified as P1, P2 or P3 on the differentials of Somé et al. [11] but (unlike the original P1, P2 or P3) are highly virulent on clubroot resistant oilseed rape cultivars.

Clubroot is characterized by the development of galls on infected roots which are often restricted in the uptake of water and nutrients and constitute a major sink for assimilates [14]. Studies on pathogen-induced changes in host metabolism and symptom development have been conducted in previous years. Ludwig-Müller et al. [15] have observed significant differences in the GSL pattern in susceptible and resistant varieties of Chinese cabbage. In their study, the total GSL content in roots of the two susceptible varieties was higher than in roots of the two resistant cultivars throughout the experimental period. While contents of aliphatic GSLs were induced in the two susceptible cultivars compared to the resistant ones, the two resistant cultivars showed an increase in aromatic GSLs, indicating maybe a dual role for these compounds. Additionally, contents of indolic GSLs (iGSLs) increased in roots of susceptible crops 14 and 20 days post inoculation with *P. brassicae*, whereas there was no difference between infected and control roots in resistant ones. Further studies on the host range of *P. brassicae* and its correlation to endogenous GSL content have shown that disease severity was correlated with certain GSLs in one species, while the increase in other GSLs might be regarded as defense response [16]. In the GSL-containing non-Brassica species, *Tropaeolum majus* and *Carica papaya*, the concentrations of benzyl-GSL increased markedly in roots inoculated with *P. brassicae*, compared with the controls. There were also increases in concentrations of benzyl-GSL in leaves of *T. majus* after *P. brassicae* infection and it was speculated that benzyl-GSL could act as precursor for phenylacetic acid that has auxin activity in *T. majus* [16].

Auxins, among them indole-3-acetic acid (IAA), are perceived by the auxin receptor transport inhibitor response 1 (TIR1). Upon binding of auxins to TIR1, the TIR1:IAA complex is formed, leading to polyubiquitination and faster degradation of already short-lived auxin responsive proteins (AUX/IAA) [17].

It has been shown that the expression of the auxin receptor *TIR1* is upregulated in *Arabidopsis thaliana* plants infected with *P. brassicae*. Furthermore, it was shown that loss of TIR1 leads to an increased susceptibility to the pathogen indicating a contribution of TIR1 to the resistance of plants towards *P. brassicae* [18].

In conclusion, degradation of GSL in general might be an important feature also during endogenous control of the clubroot disease. An improved understanding of the host metabolism and symptom development could contribute to the development of novel sources of resistance and other control strategies. Therefore, in the present study, we evaluated for the first time the variation in total and individual GSL in different clubroot-resistant and susceptible *B. napus* cultivars after inoculation with two isolates of *P. brassicae* varying in their degree of virulence.

## 2. Results

Significant differences in clubroot incidence and severity were observed between oilseed rape cultivars inoculated with different *P. brassicae* isolates at 35 dpi (Table 1). While DI and DSI in susceptible oilseed rape cultivars inoculated with *P. brassicae*-P1 were up to 100%, no disease symptoms or very small galls were observed on roots of resistant cultivars and the symptoms were of very low severity (Table 1). In contrast to *P. brassicae*-P1, the highly virulent isolate *P. brassicae*-P1 (+), had a strong effect on clubroot incidence and severity in all tested cultivars except *B. napus* cv. Creed. This cultivar was found to be completely resistant against both *P. brassicae* isolates (Table 1).

### 2.1. Individual Glucosinolate Profiles

No significant differences in GSL contents between the two runs were observed (data not shown). Data for GSL contents were therefore calculated as the mean of the two runs.

The resistant varieties Aristoteles, Mendel and Creed showed higher contents of progoitrin in leaves compared to the susceptible cultivars Bender, Ladoga and Visby, which was more pronounced in plants infected with P1 and P1 (+) compared to non-inoculated plants (Figure 1). While contents of progoitrin were lower in plants inoculated with P1 compared to non-inoculated plants, contents were similar between plants inoculated with P1 (+) and non-inoculated ones.

Similar results were obtained for glucobrassicin (I3M), although no pronounced difference was observed between non-inoculated resistant and susceptible plants. Contents were higher in inoculated plants of the resistant variety Creed compared to non-inoculated plants.

Contents of gluconapoleiferin were also slightly lower in susceptible and inoculated plants compared to the resistant varieties; however, results were not as pronounced as for the other GSLs. The resistant variety Creed had higher contents of gluconapoleiferin regardless of inoculation compared to the other two resistant varieties.

Contents of 4-methoxyglucobrassicin were lower in susceptible varieties infected with P1 and P1 (+) compared to the resistant varieties. No difference could be observed in non-inoculated plants between resistant and susceptible cultivars. Contents of resistant inoculated varieties were in the same range as contents of non-inoculated plants, whereas contents of susceptible inoculated plants were lower compared to non-inoculated plants. This trend could also be observed in all other GSLs.

Contents of neoglucobrassicin were lower in susceptible varieties infected with P1 (+) compared to plants infected with P1 and non-infected plants. Contents of resistant and inoculated plants were in the same range as contents of non-inoculated plants.

### 2.2. Mean Glucosinolate Profiles of Resistant and Susceptible Varieties

Mean contents of progoitrin in leaves were significantly lower in inoculated susceptible varieties compared to inoculated resistant varieties and non-inoculated susceptible varieties (Figure 2). Similar significant differences were also observed in the mean contents of I3M, 4-methoxyglucobrassicin, neoglucobrassicin, total aliphatic (aGSLs) and total iGSLs between resistant and susceptible varieties inoculated with P1 or P1 (+). No significant differences in mean concentration between resistant and susceptible cultivars were observed in non-inoculated plants for any of the analyzed GSLs. Mean contents of neoglucobrassicin were significantly lower in susceptible varieties inoculated with P1 (+) compared to resistant varieties. Overall, contents of all analyzed GSLs as well as total aGSLs and iGSLs were similar, independent of inoculation in resistant varieties, whereas contents in susceptible varieties were lower in plants inoculated with P1 or P1 (+) compared to non-inoculated plants.

Mean GSL contents in roots showed higher contents of iGSLs in roots compared to leaves (Figure 2 and Figure 3). Gluconasturtiin, an aGSL, was only present in roots and the most abundant GSL. Mean GSL patterns were quite similar in roots compared to leaves when looking at progoitrin and glucoalyssin, although resistant cultivars had lower contents when inoculated, compared to non-inoculated resistant plants, which cannot be observed in leaves. Contents of 4-methoxyglucobrassicin were higher in resistant as well as susceptible cultivars inoculated, compared to non-inoculated plants, which is also different to the GSL pattern in leaves, where contents in resistant inoculated plants are similar to non-inoculated ones (Figure 2).

Mean contents of progoitrin in roots of inoculated plants were lower in both resistant as well as susceptible cultivars compared to non-inoculated cultivars (Figure 3). Susceptible varieties inoculated with P1 (+) showed the lowest mean contents.

Similar results were obtained for glucoalyssin, where mean contents of inoculated plants were even lower in resistant and susceptible cultivars compared to the non-inoculated ones. Mean contents of 4-methoxyglucobrassicin were higher in inoculated resistant and susceptible plants compared to non-inoculated plants regardless of pathotype used. Mean contents of gluconasturtiin as well as neoglucobrassicin were similar in resistant cultivars regardless of inoculation, whereas susceptible varieties showed lower mean contents when inoculated compared to non-inoculated varieties.

## 3. Discussion

To our knowledge, the present work is the first study that has investigated the combined effects of virulence of the pathogen and host resistance on the variation of total and individual GSLs in roots and leaves of *Brassica napus* cultivars after inoculating with different *P. brassicae* isolates. The results clearly showed that clubroot severity depended significantly on the virulence of the pathogen and the susceptibility of the oilseed rape cultivars (Table 1). Successful inoculation is reflected in the disease incidence and severity, which is low to non-existent in resistant cultivars and very high in susceptible varieties (Table 1). Only low disease incidences for the resistant cultivar Mendel inoculated with P1, and low disease severity, and slightly higher incidences and disease severities for the cultivars Mendel and Aristoteles inoculated with P1 (+) highlight the virulence selection of the *P. brassicae* isolates. Very high disease incidents and severities for susceptible plants inoculated with either P1 or P1 (+) show their suitability for this research question because of their high susceptibility towards the pathogen.

### 3.1. Similarity of Glucosinolate Contents in Resistant Cultivars—A Coincidence?

Genome mapping of *Brassica rapa* var. *rapifera* as well as *B. oleraceae* var. *pekinensis* revealed 11 loci that convey resistance to *P. brassicae* [19]. The resistant cultivar Mendel is known to possess five of them, among them *CRa*, which is known to encode for a TIR-NB-LRR class disease-resistance protein [19,20]. This protein consists of a toll interleukin 1 receptor domain, involved in pathogen recognition, a nucleotide binding (NB) domain, involved in disease signaling pathways and a leucine rich repeat (LRR) domain, which is responsible for protein-protein interactions and ligand binding. Because of its distinct function involved in pathogen detection it is unlikely involved in the synthesis or regulation of GSLs [21]. Unfortunately, the function of the other loci responsible for the resistance are not known and no information about the presence of these loci in the resistant cultivars Aristoteles and Creed is available.

Nevertheless, it is noteworthy that all resistant cultivars showed similar GSL contents in leaves (Figure 1 and Figure 2) as well as in roots (Figure 3). The similar GSL contents highlight a general effect of the infection on the resistant cultivars despite putative differing genetic backgrounds. This phenomenon could be attributed to normal GSL biosynthesis and breakdown in resistant cultivars compared to susceptible cultivars because the pathogen might not be able to interfere with GSL synthesis or degradation, which is reflected in the non-existent to low disease incidence and severity of resistant cultivars (Table 1).

### 3.2. Higher Contents of Indolic Glucosinolates—A Double-Edged Sword

Resistant inoculated cultivars showed higher contents of the iGSLs I3M, 4-methoxglucobrassicin and neoglucobrassicin in leaves compared to susceptible inoculated varieties (Figure 1). Results became more pronounced when looking at the mean total contents of iGSLs (Figure 2). Glucosinolates and classical thioglucosidases (EC 3.2.1.147) are usually stored in different cell compartments or different cells and come together upon tissue disruption resulting in degradation of GSLs. A Glu residue in the catalytic site of classical thioglucosidases performs a nucleophilic attack on GSLs resulting in an aglucone. Ascorbic acid is then needed as proton donor to cleave the glucose from the aglucone [22].

However, GSLs can also be degraded in intact tissue in the presence of atypical thioglucosidases like PEN2 and PYK10 [23,24]. Two Glu residues in the active sites of atypical thioglucosidases perform an acid/base catalysis, which does not require ascorbic acid. In contrast to typical thioglucosidases, atypical thioglucosidases also accept O-glucosides alongside GSLs as substrates [22]. Upon degradation of I3M with thioglucosidases and the presence of nitrile specifier proteins, indole-3-acetonitrile (IAN) can be formed and further synthesized to auxin in the presence of nitrilases (EC 3.5.5.1) [25]. Higher contents of auxin were found to be responsible for cell elongation in roots, susceptibility and subsequent gall formation induced by an infection with *P. brassicae* [21].

Nevertheless, indole-3-acetic acid (IAA) is not the only outcome of the breakdown of I3M. If nitrile specifier proteins (NSP1, NSP5) are not present during degradation of I3M, indolylmethylisothiocyanate is formed and quickly reacts to indole-3-carbinol, which usually conjugates with nucleophilic compounds like cysteine, glutathione or ascorbic acid [26]. Molecular docking simulations revealed that some of these conjugates possess potent auxin inhibitory activities by binding to the auxin receptor TIR1 and blocking the subsequent binding of IAA and its interaction with auxin regulatory proteins (AUX/IAA [17]. It was also hypothesized that during the infection with a pathogen, the synthesis of I3M breakdown products (BP) could lead to a drop of TIR1:IAA complex levels, and therefore, uncoupling of IAA perception from actual IAA contents takes place leading to normalized IAA signaling [17] (Figure 4c). Inhibition of auxin perception in *A. thaliana* was shown with the addition of indole-3-carbinol to growth media, which was able to inhibit root elongation in a concentration-dependent manner [27]. Due to its root-growth repressive action, this compound might be involved in the inhibition of gall growth directly in the roots by blocking TIR1:IAA formation and subsequent auxin signaling.

Interestingly, in our study contents of iGSLs in leaves were unchanged upon infection with *P. brassicae* in resistant cultivars compared to non-inoculated plants, whereas contents were lower in susceptible inoculated plants compared to non-inoculated plants (Figure 1). In order to avoid auxin-inhibitory actions of I3M BP, the pathogen might influence biosynthesis of iGSLs, which is reflected in lower levels of I3M in leaves of susceptible cultivars. Slightly lower contents of I3M in roots of inoculated susceptible plants compared to resistant ones (Figure 3) also points in this direction. The lower contents in roots could lead to a positive feedback of auxin due to degradation of inhibitory AUX/IAA proteins by the formation of the TIR1:IAA complexes (Figure 4d).

Contents of neoglucobrassicin and 4-methoxyglucobrassicin were also lower in leaves of susceptible plants inoculated with P1 and P1 (+) (Figure 1 and Figure 2) and roots of susceptible plants inoculated with P1 (Figure 3) compared to resistant ones. Since the properties of the BPs of 4-methoxyglucobrassicin and neoglucobrassicin are not known at this point, only speculations can be made about their inhibitory activity towards the formation of the TIR1:IAA complex. Nevertheless, it is important to keep in mind that *P. brassicae* could also interfere with the expression of genes or synthesis of proteins involved in the degradation of GSLs and modification of BPs, which would not be reflected in GSLs contents. It was shown in *A. thaliana* that both leaves and roots are capable of synthesizing iGSLs and that GSL transporters are in charge of the long-distance transport of GSLs, which raises the question why differences between resistant and susceptible cultivars are more pronounced in leaves compared to roots [28].

### 3.3. Is There an Interplay Between Aliphatic and Indolic Glucosinolates?

Single (Figure 1) as well as mean total aGSL contents (Figure 2) were observed to be reduced in susceptible cultivars upon infection with *P. brassicae*. On the other hand, contents in inoculated resistant cultivars remain in the same range compared to non-inoculated ones (Figure 1 and Figure 2). Maintenance of water balance is of importance in plants infected with *P. brassicae* since the pathogen disrupts water uptake by the host [29]. Sustention of aGSL contents in resistant cultivars during an infection with *P. brassicae* might be beneficial due to the involvement of BPs derived from aGSLs in stomatal aperture, which could lead to contained water loss. It has been shown that allylisothiocyanate and 3-butenenitril, both BPs of sinigrin, as well as ethylthiocyanate, a BP of glucolepidiin, lead to stomatal closure through generation of reactive oxygen species (ROS) which was reversed by addition of catalase [30,31]. It is possible, that the degradation products of other aGSLs, like progoitrin present in *B. napus* (Figure 1 and Figure 2), could also trigger closure of stomata.

It has been shown that the auxin responsive proteins IAA5, IAA6 and IAA19 repress the expression of *WRKY63*, which encodes for a transcription factor inhibiting the expression of *MYB28/29*. The transcription factors MYP28/29 are involved in the positive regulation of aGSL synthesis [32]. Lower expression of *WRKY63*, negatively influencing aGSL contents, mediated by IAA5, IAA6 and IAA19, therefore leads to higher aGSL contents. Breakdown of auxin-responsive proteins is usually mediated by TIR1:IAA complex formation, but can be inhibited by docking of I3M BPs to TIR1 [17]. The prolonged lifespan of IAA5, IAA6 and IAA19 could therefore lead to higher contents of aGSLs via repression of the TIR1:IAA formation, mediated by the breakdown of iGSLs. Higher contents of aGSLs and their subsequent degradation might be used by the plant to attenuate drought symptoms caused by *P. brassicae* through closure of stomata (Figure 4a). On the other hand, the synthesis of TIR1:IAA inhibitory compounds could be inhibited in susceptible cultivars by the pathogen, resulting in faster degradation of IAA5, IAA6 and IAA19 and therefore lower contents of aGSLs (Figure 4b). This could also explain the lower contents of aGSLs in susceptible cultivars infected with P1 (+) compared to P1. This data suggests that virulence of the isolates might correlate with the degree of interference with the plants’ metabolism.

### 3.4. Direct Effect of Breakdown Products on P. brassicae

Levels of gluconasturtiin in roots were observed to be lower in susceptible varieties inoculated with P1 and even lower in plants inoculated with P1 (+) compared to resistant varieties (Figure 3). Analysis of quantitative trait loci involved in resistance and metabolic changes revealed a possible involvement of gluconasturtiin in the infection with *P. brassicae* [33]. The higher levels of gluconasturtiin controlled by resistance alleles found in the mentioned publication is in accordance with the findings of this study. As gluconasturtiin is the most abundant GSL in roots of the chosen cultivars analyzed, the contents of which are unchanged in resistant cultivars upon an infection, the pathogen might interfere with biosynthesis of gluconasturtiin in susceptible varieties. The different isolates of *P. brassicae* analyzed seem to differ in their ability to reduce gluconasturtiin contents in the host. The more aggressive isolate P1 (+) might therefore be able to suppress gluconasturtiin synthesis of the host in a more pronounced manner compared to the isolate P1. At this moment, only assumptions can be made about the effects of gluconasturtiin on the pathogen, although direct effects of the isothiocyanate derived from this GSL on *P. brassicae* might be more likely.

## 4. Materials and Methods

### 4.1. Plant and Pathogen Materials

Six oilseed rape cultivars (*Brassica napus* L.) with a different resistance level to clubroot disease were selected in the current study according to the German Plant Variety Catalogue in 2018 (Table 2). These cultivars were pre-selected according to results obtained from pre-experiments (data not shown). Selected resistant cultivars had no to low disease symptoms and selected susceptible cultivars had a high disease incidence as well as disease severity and were therefore selected for this study. Although information about the genes responsible for the resistance towards *Plasmodiophora brassicae* is only available for the cultivar Mendel, the aim of this study was to observe general changes in GSLs despite putatively different genetic backgrounds.

Two field isolates of *P. brassicae* were chosen according to their evaluated virulence in Zamani-Noor (2017) [9]: an isolate which was virulent on clubroot susceptible cv. Visby, avirulent on clubroot resistant cv. Mendel, classified as 16/31/12 on the European clubroot differential (ECD) set [12] or pathotype 1 according to the system of Somé et al. [11] (briefly named P1), and a highly virulent isolate which could overcome the resistance of both cultivars, classified as 17/31/31 on the ECD set or pathotype 1 in system of Somé et al. [11] (briefly named P1 (+)). The P1 isolate was collected from a naturally infested oilseed rape field in Hoisdorf, Schleswig-Holstein, Germany, in 2012 and the P1 (+) isolate originated from a field in Grävenwiesbach, Hesse, Germany, in 2013 [9]. Both isolates were preserved as frozen root galls in −20 °C and used for inoculum preparation as needed.

### 4.2. Plant Cultivation and Inoculation

The experiments were conducted under controlled greenhouse conditions using portable raised-bed containers (300 × 100 × 25 cm) containing a mix of potting soil, sand and peat (5:1:1; pH < 6.5; FloraSelf^®^, Braunschweig, Germany). Seeds were sown at 7.5 cm spacing in a row spaced 11.5 cm apart from the other row. In total, 17–20 seeds per row were sown and seedlings were thinned on emergence to leave 13 plants, and there were four rows per oilseed rape cultivar. Plants were grown under greenhouse conditions at 20/16 °C, 70% relative humidity and a 16/8 h day/night regime with a light intensity of 50 µmol m^−2^ s^−1^. Plants were inoculated at growth stage 11–12 (BBCH-scale; young seedling) and were well irrigated prior to inoculation.

The inoculum was prepared according to in Zamani-Noor (2017) [9]. In summary, the resting spores of each *P. brassicae* isolate were released from frozen clubbed roots by homogenizing 100 g clubbed roots in 200 mL of sterile deionized water in a laboratory blender for 5 min at 20,000 rpm (Vital Mixer Pro, Hollenstedt, Germany). The solution was filtered several times through fine layers of cheesecloth until the suspension was free from plant debris. The spore suspension was diluted to a concentration of 1 × 10^7^ spores per mL as estimated using a Fuchs haemocytometer slide (Hecht-Assistent, Sondheim, Germany) under a microscope.

Inoculations were conducted by injecting 2 × 1 mL of spore suspension (1 × 10^7^ spores per mL) into the soil at two locations near the root zone of each seedling at a depth of approximately 2 cm. Control plants were mock inoculated in the same way with water. To avoid washing the inoculum from the root area, the plants were not irrigated for 72 h post inoculation and were kept at a temperature of 24 °C to attain the best conditions for the infection. Following this period, plants were grown at previous greenhouse conditions and irrigated every other day to maintain soil moisture, but they were not water saturated.

### 4.3. Plant Sampling and Disease Assessment

Roots and leaves samples were collected one day before inoculation (plant growth stage BBCH 11-12) and on 35 days post inoculation (dpi). At each date, nine plants per oilseed rape cultivar were completely dug out and divided into 3 biological replicates consisting of three plants each. Leaves were separated and frozen immediately in liquid nitrogen and then stored in −80 °C for further steps. The roots were then carefully washed under tap water to remove soil particles and clubroot severity was visually assessed based on a scale of 0 to 3 (0 = no galling, 1 = a few small galls, 2 = moderate galling and 3 = severe galling) [34]. Conclusively, roots were frozen in liquid nitrogen and then stored at −80 °C.

The disease incidence (DI) and disease severity index (DSI) were calculated for each treatment using Equations (1) and (2):(1)$$\mathrm{DI}\;\left(\%\right)=\frac{\sum \left(n1+n2+n3\right)}{N}\times 100$$
(2)$$\mathrm{DSI}\;\left(\%\right)=\frac{\sum \left(n0\times 0+n1\times 1+n2\times 2+n3\times 3\right)}{N\times \mathrm{No}.\mathrm{Classes}\;\mathrm{with}\;\mathrm{symptoms}}\times 100$$
where ‘*n*’ is the number of plants in each class, ‘*N*’ is the total number of plants and values 0, 1, 2 and 3 represent the respective symptom severity classes.

### 4.4. Extraction of Glucosinolates

Samples were prepared as described by Hornbacher et al. [35]. Briefly, frozen plant materials were lyophilized in a freeze dryer (Martin Christ Gefriertrocknungsanlagen GmbH, Osterode am Harz, Germany) for 2 days and ground to a fine powder with a shaking ball mill (Retsch GmbH, Braunschweig, Germany). Approximately 50 mg dry plant tissue was extracted with 1 mL 80% methanol at room temperature for 10 min and then centrifuged at 13,000× *g* for 5 min. Before the centrifugation, samples were put on a shaker for 15 min after the first extraction and 30 min after the second extraction at room temperature (RT). The supernatants were pooled and loaded onto a column (polypropylene column, 1 mL) containing 2 mL of a 5% (*w*/*v*) suspension of DEAE Sephadex A25 (Sigma-Aldrich, Taufkirchen, Germany) in 0.5 M acetic acid (pH 5). Columns were washed five times with 2 mL H_2_O and two times with 2 mL 0.02 M acetic acid (pH 5). For desulfation, 50 μL of sulfatase (Sigma-Aldrich, Taufkirchen, Germany) solution was added to 450 µL 0.02 M acetic acid (pH 5) and loaded onto the columns as well [36]. Desulfation took place for 24 h at RT. Afterwards desulfated GSLs were eluted three times with 2 mL HPLC H_2_O (Sigma-Aldrich, Taufkirchen, Germany), dried overnight in a vacuum centrifuge and then dissolved in a total amount of 300 μL HPLC H_2_O.

### 4.5. Liquid Chromatography Mass Spectrometry (LCMS) and Analysis of Glucosinolates

Glucosinolate contents in oilseed rape samples were analyzed via liquid chromatography–mass spectrometry (LC-MS). A volume of 10 μL was injected into the HPLC system (Shimadzu, Darmstadt, Germany) and separated on a Knauer Vertex Plus column (250 × 4 mm, 5 μm particle size, packing material ProntoSIL 120-5 C18-H) equipped with a pre-column (Knauer, Berlin, Germany). A water (solvent A)-methanol (solvent B), both containing 2 mM ammonium acetate, gradient was used with a flow rate of 0.8 mL min at 30 °C. For measuring the samples, the following gradient was used: 10–90% B for 35 min, 90% for 2 min, 90–10% B for 1 min and 10% B for 2 min. Detection of the spectra in the range 190–800 nm was performed with a diode array detector (SPD-M20A, Shimadzu, Darmstadt, Germany). The HPLC system was coupled to an AB Sciex Triple TOF mass spectrometer (AB Sciex TripleTOF 4600, Canby, OR, USA). At a temperature of 600 °C and an ion spray voltage floating of −4500 V the negative electrospray ionization (ESI) was performed. For the ion source gas one and two 50 psi were used and for the curtain gas 35 psi. In the range of 100–1500 Da in the TOF range, the mass spectra as well as the MS/MS spectra from 150–1500 Da at a collision energy of −10 eV were recorded. Peaks were identified by analyzing the characteristic mass fragments of ds-progoitrin (195, 309, 344, 617), ds-glucoalyssin (195, 208, 371, 741) and ds-neoglucobrassicin (195, 208, 371, 741). The detection of the GSL was performed with DAD (Knauer, Berlin, Germany) at 229 nm. Quantification of the measured GSL was performed using sinigrin (Phytolab, Vestenbergsgreuth, Germany) as external standard and relative response factors (progoitrin, 1.09; glucoraphanin, 1.07; glucoalyssin, 1.07; gluconapin, 1.11; hydroxyglucobrassicin, 0.17; glucobrassicanapin, 1.15; I3M, 0.29; gluconasturtiin, 0.95; neoglucobrassicin, 0.2). Integration of peaks and elaboration of data were performed using PeakView software version 2.1.0.1 (AB Sciex, Darmstadt, Germany). Limits of quantification for aGSL were determined with glucoraphanin as standard (Phytolab, Vestenbergsgreuth, Germany) and for iGSL I3M (Phytolab, Vestenbergsgreuth, Germany) was used. Limits of quantification were determined to be 30 nmol mL^−1^ for aGSLs and 6 nmol mL^−1^ for iGSLs. Glucosinolate contents (total GSL, aGSLs, iGSLs and aromatic GSLs) were calculated as the mean of three biological replicates, consisting of three plants each, with standard deviation of the three replicates. The total amount of GSLs for each sample was calculated as the sum of all individual GSLs. Total aGSL contents were calculated as the sum of the contents of progoitrin and gluconapoleiferin. Total iGSL contents were calculated as the sum of I3M, 4-methoxyglucobrassicin and neoglucobrassicin. Although contents of gluconapin and glucobrassicanapin were analyzed, they are not shown due to very low levels and high standard deviations.

### 4.6. Statistical Analysis

All experiments in the present study were conducted twice, where each repetition is referred to as a run. With regards to different treatments or independent factors in our experiments, we decided to use analysis of variance (ANOVA) over multiple *t*-test. Analysis of common treatments did not show significant differences (*p* ≤ 0.05) between two runs, so we pooled the data for analysis and presentation in this study.

Glucosinolate concentrations were log-transformed before analysis due to finding right skewed distributions and variance increasing with mean. For the log-transformed data, linear mixed effect models were fitted to account for the split-plot design. The experimental runs, the pathogens, the traits and the varieties nested within the traits were included as fixed effects. For the latter three, the corresponding interactions were also modelled. Furthermore, three random effects were included: the main plot, which accounts for the variance of the spatial separation between the pathogen inoculations; the subplot, which takes into consideration the variance of the varieties within the main plot; and a random effect, which represents the variance of the varieties between the two experiments. Based on the fitted linear mixed models, analysis of variance was performed to test the significance of main effects and interactions and Tukey tests for the model-based means of factor variety were performed jointly across all other factor levels and separately for inoculated and non-inoculated groups.

## 5. Conclusions

Although it was previously hypothesized that loss of iGSLs has no influence on gall formation in *Arabidopsis thaliana,* our results and hypotheses are quite in line with the findings of Siemens et al. [14]. The mutants used by Siemens et al. [14] (*cyp79b2/b3*) were incapable of synthesizing iGSLs and showed no difference in gall formation compared to wild type plants. The loss of iGSLs also leads to a loss of a variety of BPs that could inhibit TIR1:IAA formation and therefore stabilize auxin signaling during an infection with *P. brassicae*.

It will be necessary to perform gene expression analysis as well as further analysis of GSL contents because observed changes in GSL contents could be attributed to the infection itself as a correlated symptom without having a causal relationship.

Follow-up experiments could focus on *nsp* mutants which would be incapable of directing the outcome of I3M degradation towards IAN, and these plants would be left with higher contents of I3M conjugates and therefore a possible uncoupling of auxin signaling would take place.

Expression analysis of *TIR1* as well as *AUX/IAA* and *NSP1,5* would also help to substantiate the hypotheses made in this work. Analysis of free as well as conjugated auxin would also be crucial for the understanding of the involvement of iGSLs during an infection with *P. brassicae*.

Finally, analysis of GSLs as well as expression analysis at different time points during the infection would allow a more distinct insight into the time-dependent actions of *P. brassicae* in *B. napus.*

## Figures and Tables

**Figure 1 pathogens-10-00563-f001:**
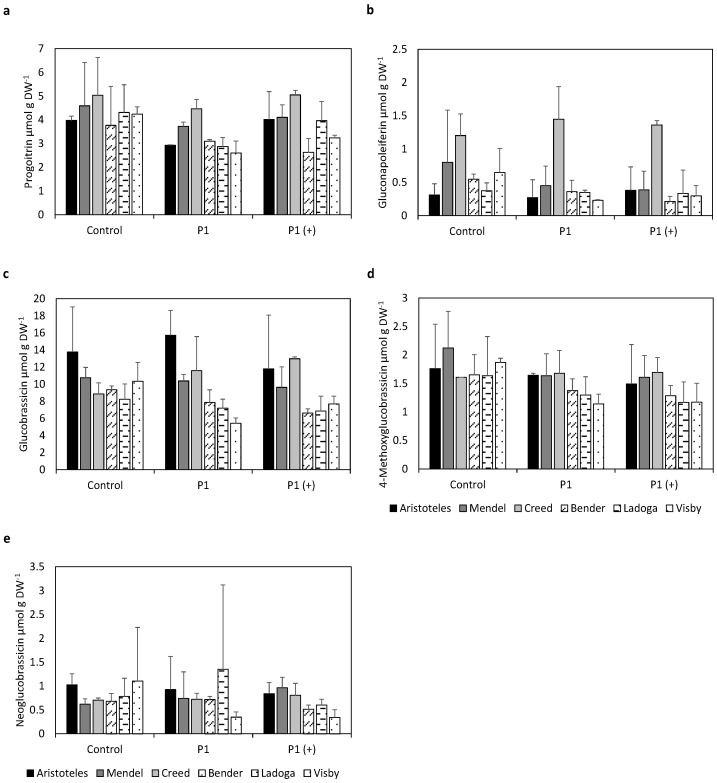
Contents of (**a**) Progoitrin, (**b**) Gluconapoleiferin, (**c**) Glucobrassicin, (**d**) 4-Methoxyglucobrassicin and (**e**) Neoglucobrassicin in leaves of clubroot-resistant (Aristoteles, Mendel, Creed) and -susceptible (Bender, Ladoga, Visby) cultivars of *Brassica napus* 35 days post inoculation (dpi) after mock inoculation (control) or artificial inoculation with either a less virulent isolate (P1, Hoisdorf) or a more virulent isolate (P1 (+), Grävenwiesbach) of *Plasmodiophora brassicae*. Contents represent the mean of six biological replicates, consisting of three plants each, from two independent experiments. Error bars represent the standard deviation. An analysis of variance (ANOVA) was performed to test the significance of main effects and interactions (for *p*-values see Appendix A). No significant differences were observed. DW, dry weight.

**Figure 2 pathogens-10-00563-f002:**
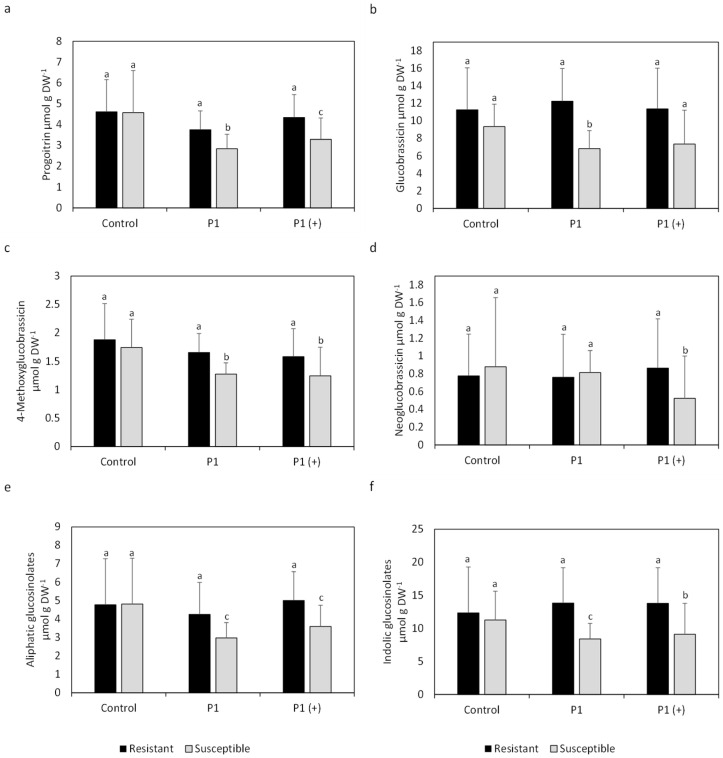
Contents of (**a**) Progoitrin, (**b**) Glucobrassicin, (**c**) 4-Methoxyglucobrassicin, (**d**) Neoglucobrassicin, (**e**) total aGSLs and (**f**) total iGSLs in leaves of clubroot-resistant (mean of cultivars Aristoteles, Mendel, Creed) and -susceptible (mean of cultivars Bender, Ladoga, Visby) cultivars of *Brassica napus* 35 days post inoculation (dpi) after mock inoculation (control) or inoculation with either a less virulent isolate (P1, Hoisdorf) or a more virulent isolate (P1 (+), Grävenwiesbach) of *Plasmodiophora brassicae*. Contents represent the mean of six biological replicates, consisting of three plants each, from two independent experiments. Error bars represent the standard deviation. An analysis of variance (ANOVA) was performed to test the significance of main effects and interactions, and Tukey’s post hoc tests for the means of the factor variety were performed separately for inoculated and non-inoculated groups (for *p*-values see Appendix A). Different letters present significant differences between resistant and susceptible cultivars (a = *p* > 0.05; b = *p* < 0.05; c = *p* < 0.005). No significant differences were observed for other glucosinolates. DW, dry weight.

**Figure 3 pathogens-10-00563-f003:**
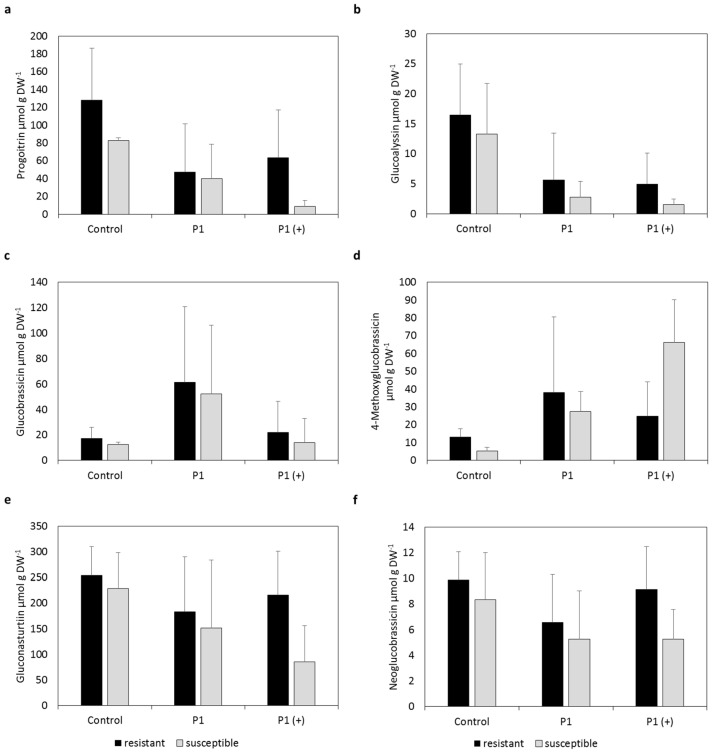
Contents of (**a**) Progoitrin, (**b**) Glucoalyssin, (**c**) Glucobrassicin, (**d**) 4-Methoxyglucobrassicin, (**e**) Gluconasturtiin and (**f**) Neoglucobrassicin in roots of clubroot-resistant (mean of cultivars Aristoteles, Mendel, Creed) and -susceptible (mean of cultivars Bender, Ladoga, Visby) cultivars of *Brassica napus* 35 days post inoculation (dpi) after mock inoculation (control) or inoculation with either a less virulent isolate (P1, Hoisdorf) or a more virulent isolate (P1 (+), Grävenwiesbach) of *Plasmodiophora brassicae*. Contents represent the mean of six biological replicates, consisting of three plants each, from two independent experiments. Error bars represent the standard deviation. An analysis of variance (ANOVA) was performed to test the significance of main effects and interactions, and Tukey’s post hoc tests for the means of the factor variety were performed separately for inoculated and non-inoculated groups (for *p*-values see Appendix A). No significant differences were observed. DW, dry weight.

**Figure 4 pathogens-10-00563-f004:**
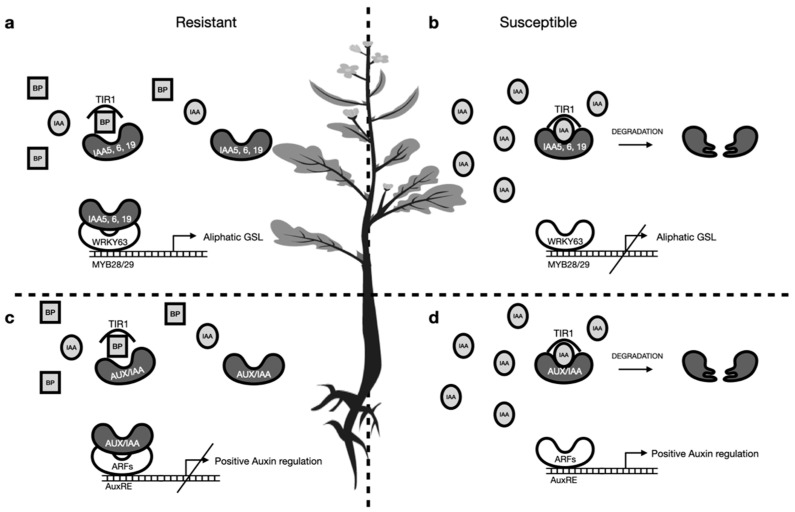
Proposed model for the role of iGSLs upon an infection with *Plasmodiophora brassicae* in *Brassica napus* and interplay between iGSLs and aGSLs. (**a**) Resistant cultivars might maintain the water status in leaves by involving breakdown products of aGSLs in stomatal closure. Upregulation of aGSL contents was shown to be mediated by IAA5, IAA6 and IAA19. Stability of these auxin-responsive proteins might be prolonged by inhibition of the TIR1 receptor with conjugated I3M breakdown products. (**b**) Susceptible cultivars might be restrained in their production of I3M breakdown products due to low iGSL contents. This might lead to degradation of IAA5, IAA6 and IAA19 through IAA and subsequent drop in contents of aGSLs. (**c**) Inhibition of the TIR1 receptor by I3M breakdown products in resistant cultivars might maintain functional auxin response despite higher IAA contents during an infection. (**d**) Lower levels of iGSLs and subsequent lower levels of I3M breakdown products might lead to a positive feedback response caused by high auxin concentrations. Legend: IAA = indole-3-acetic acid; BP = breakdown products of glucobrassicin; TIR1 = transport inhibitor response 1, IAA receptor; AUX/IAA = auxin responsive proteins, mostly IAA repressive proteins; IAA5, 6, 19 = auxin responsive proteins, inhibit WRKY63; WRKY63 = transcription factor, represses expression of MYB28/29; MYB28/29 = transcription factors, positively regulate biosynthetic genes in the synthesis of aGSLs.

**Table 1 pathogens-10-00563-t001:** Clubroot disease incidence (DI) and disease severity index (DSI) of different clubroot resistant and susceptible oilseed rape (*Brassica napus*) cultivars inoculated with two isolates of *Plasmodiophora brassicae* at 35 days post inoculation (dpi).

Cultivar	Clubroot Resistance	*P. brassicae* P1 ^1^		*P. brassicae* P1 (+) ^1^	
		DI ^2^ ± SD	DSI ^2^ ± SD	DI ^2^ ± SD	DSI ^2^ ± SD
Aristoteles	resistant	0.0 ± 0.0	0.0 ± 0.0	44.0 ± 3.0	23.3 ± 1.6
Creed	resistant	0.0 ± 0.0	0.0 ± 0.0	0.0 ± 0.0	0.0 ± 0.0
Mendel	resistant	15.3 ± 12.6	9.9 ± 6.8	60.0 ± 8.3	30.5 ± 3.6
Bender	susceptible	100.0 ± 0.0	100.0 ± 0.0	100.0 ± 0.0	98.8 ± 1.7
Ladoga	susceptible	100.0 ± 0.0	98.2 ± 2.5	97.9 ± 3.0	91.3 ± 12.6
Visby	susceptible	100.0 ± 0.0	98.8 ± 1.9	100.0 ± 0.0	100.0 ± 0.0

^1^*P. brassicae* isolates were chosen according to their evaluated virulence in Zamani-Noor (2017) [9]. Pb-P1 isolate was virulent on clubroot susceptible oilseed rape cv. Visby and avirulent on clubroot resistant cv. Mendel and Pb-P1 (+) isolate was virulent on both cultivars. ^2^ The infection type on each root was visually determined based on a 0–3 scale; disease incidence (DI) and disease severity index (DSI) were calculated from each infection type. Data are pooled across two experimental runs (i.e., repetitions); mean values and standard deviations (± SD) are presented in this table.

**Table 2 pathogens-10-00563-t002:** Cultivars of *Brassica napus* and their level of resistance to clubroot disease caused by *Plasmodiophora brassicae.*

Cultivar	Seed Source	Clubroot Resistance
Aristoteles	Limagrain GmbH	resistance: single dominant gene (based on ‘Mendel’ resistance)
Creed	Norddeutsche Pflanzenzucht	resistance: polygenic resistance (internal communication with the company)
Mendel	Norddeutsche Pflanzenzucht	resistance: single dominant gene-based resistance [5]
Bender	Deutsche Saatveredelung AG	susceptible
Ladoga	Limagrain GmbH	susceptible
Visby	Norddeutsche Pflanzenzucht	susceptible

## Data Availability

Data presented in this study are available upon request.

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
