# Peer review of "Variation of Glucosinolate Contents in Clubroot-Resistant and -Susceptible Brassica napus Cultivars in Response to Virulence of Plasmodiophora brassicae"

_pathogens, 2021, doi:10.3390/pathogens10050563_

Round 1

Reviewer 1 Report

Comments: This study mainly investigated the individual and total glucosinolate contents in resistant and susceptible cultivars inoculated with P1 and P1(+), which will provide useful information to the understanding of host-pathogen interactions.

A good research.

Reviewer 3 Report

I found this manuscript very interesting, it brings new knowledge about mechanisms of pathogen-host mechanisms based on glucosinolates content and changes. I don't have many comments. The manuscript is very well arranged and written. There are just a few typos:

line 225 - remaining comma between pathogen detection

line 259 - remaining reference with the name of the author

In Material and Methods, I would suggest filling in some references in the part about the preparation of spore suspension and plant inoculation.

Round 2
